**Genomic Data Analyses in Biobanks**

# Mendelian randomization with proxy exposures: challenges and opportunities

Ida Rahu [ID],[1,2] Ralf Tambets,[1] Eric B. Fauman,[3] Kaur Alasoo [ID] [1,*]

[1]Institute of Computer Science, University of Tartu, Tartu 51009, Estonia
[2]Department of Environmental Science, Stockholm University, Stockholm 114 18, Sweden
[3]Research and Development, Internal Medicine Research Unit, Pfizer, Cambridge, MA 02139, United States

*Corresponding author: Institute of Computer Science, University of Tartu, Tartu 51009, Estonia. Email: kaur.alasoo@ut.ee

A key challenge in human genetics is the discovery of modifiable causal risk factors for complex traits and diseases. Mendelian randomization (MR) using molecular traits as exposures is a particularly promising approach for identifying such risk factors. Despite early successes with the application of MR to biomarkers such as low-density lipoprotein cholesterol and C-reactive protein, recent studies have revealed a more nuanced picture, with widespread horizontal pleiotropy. Using data from the UK Biobank, we illustrate the issue of horizontal pleiotropy with 2 case studies, one involving glycolysis and the other involving vitamin D synthesis. We demonstrate that, although the measured metabolites (pyruvate or histidine, respectively) do not have a direct causal effect on the outcomes of interest (red blood cell count or vitamin D level), we can still use variant effects on these downstream metabolites to infer how they perturb protein function in different gene regions. This allows us to use variant effects on metabolite levels as proxy exposures in a *cis*-MR framework, thus rediscovering the causal roles of histidine ammonia lyase (*HAL*) in vitamin D synthesis and glycolysis pathway in red blood cell survival. We also highlight the assumptions that need to be satisfied for *cis*-MR with proxy exposures to yield valid inferences and discuss the practical challenges of meeting these assumptions.

Keywords: Mendelian randomization; pleiotropy; UK Biobank; HAL; vitamin D; glycolysis

## Introduction

A key challenge in human genetics is identifying modifiable causal risk factors for complex traits and distinguishing those from other biomarkers with no causal effect. For example, many cardiovascular disease loci are also associated with low-density lipoprotein (LDL) cholesterol level, a known causal risk factor for cardiovascular disease (Ference et al. 2012; Richardson et al. 2022). Furthermore, Mendelian randomization (MR) studies have demonstrated that individuals with a genetic predisposition to lower LDL cholesterol level also have a reduced risk of cardiovascular disease (Voight et al. 2012). This link has been confirmed by clinical trials demonstrating the success of lipid-lowering therapies in reducing cardiovascular disease risk (Mihaylova et al. 2024). In contrast, results from MR studies are not consistent with a causal link between C-reactive protein (CRP) and cardiovascular disease, despite a strong observational correlation (C Reactive Protein Coronary Heart Disease Genetics Collaboration (CCGC) et al. 2011).

The hope of replicating the success in identifying modifiable biomarkers such as LDL cholesterol for other traits has prompted the high-throughput measurements of thousands of accessible molecular traits from tens to hundreds of thousands of individuals in existing large biobanks. These molecular traits include plasma metabolites (Richardson et al. 2022; Smith et al. 2022; Karjalainen et al. 2024), plasma proteomics (Sun et al. 2018, 2023), and transcriptomic data from whole blood (Võsa et al. 2021). Relying on easily accessible whole blood and plasma samples has enabled these studies to attain sufficient sample sizes to capture associations with low-frequency variants, as well as genetic associations with small effects. As a result, these studies now routinely identify thousands of associations. Furthermore, while early proteomic and transcriptomic studies focused on genetic variants located near the protein-coding genes to map *cis*-quantitative trait loci (*cis*-QTLs; Fig. 1a), increased sample sizes mean that most detected associations are now located in *trans* and affect the target gene or protein levels via the activity of *trans*-acting factors (typically other proteins; Fig. 1c) (Sun et al. 2023). These genetic resources provide a large number of genetic instruments for MR studies, contributing to the rapid increase in MR studies in the literature (Richmond and Davey Smith 2022; Sanderson et al. 2022; Stender et al. 2024).

However, inferences from MR studies are only valid if certain assumptions are met (Burgess et al. 2019; Skrivankova et al. 2021). A key assumption of MR is that the genetic variants are associated with the outcome only via the exposure of interest (Reed et al. 2025). This assumption can be violated by *horizontal pleiotropy*, where the causal effect of the genetic variants on the outcome is mediated by another trait not included in the analysis (Sanderson et al. 2024). Importantly, genetic instruments identified for high-throughput protein, transcript, or metabolite measurements are often subject to horizontal pleiotropy, leading to incorrect or misleading MR inferences (Richardson et al. 2022; Smith et al. 2022; Karjalainen et al. 2024) (Fig. 1). As an example,

**a** Horizontal pleiotropy due to cell-type-specifc gene regulation

**b** Horizontal pleiotropy due to local co-regulation in *cis*

**c** Horizontal pleiotropy due to co-regulation in *trans*-QTL networks

**d** Using *trans*-QTL effects as proxy masures for gene A function avoids horizontal pleiotropy

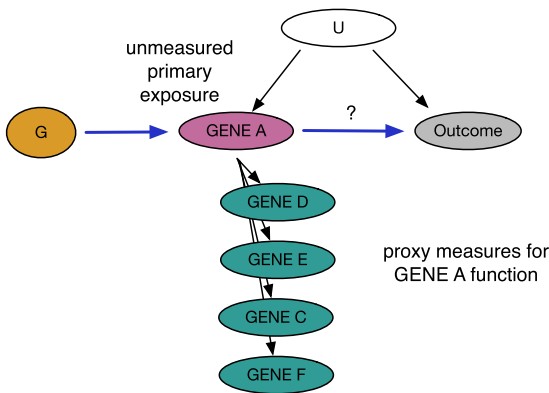

**Fig. 1.** Some molecular mechanisms of horizontal pleiotropy. a) In the presence of cell-type-specific QTLs, the same gene profiled in a different cell type can be a source of horizontal pleiotropy. Note that cell-type-specific QTLs G1 and G2 could be in linkage disequilibrium (LD) with each other (dashed line), further complicating the inference. b) Horizontal pleiotropy due to local coregulation of gene expression. Example of a *cis*-QTL variant G that is associated with the expression of gene A (primary exposure) and also with a neighboring gene B (alternative phenotype). In the causal diagram, potential horizontal pleiotropy is limited to a small number of locally coregulated genes. c) Horizontal pleiotropy due to coregulation in *trans*-QTL networks. Example of a *trans*-QTL variant G that is associated with the expression of genes C–F on different chromosomes. Note that the *trans*-QTL effect of variant G on genes C–F is typically mediated by at least 1 gene in the *cis*-region (eg gene A). High degree of horizontal pleiotropy can make it challenging to identify the true causal mediators. The variant effect on *cis*-gene A function is treated as an unmeasured intermediate phenotype (vertical pleiotropy). d) The same scenario as in c), but now the exposure of interest is gene A function, which is proxied by the variant effect on downstream genes C–F. Horizontal pleiotropy can be avoided in the absence of cell-type-specific regulatory effects (a) or in the absence of local coregulation in the *cis*-region (b). G, genetic instruments; U, unmeasured confounders.

Karjalainen et al. reported that MR between acetone and 233 other metabolites identified 20 significant associations, mostly with lipid traits, but almost all these associations were attenuated when pleiotropic variants at well-known lipid loci were excluded (Karjalainen et al. 2024). Restricting the analysis to 4 less pleiotropic instruments identified a putative causal association between plasma acetone level and hypertension (Karjalainen et al. 2024). Similarly, both proteomic and transcriptomic studies have identified pleiotropic regulatory variants associated with the abundance of tens to hundreds of genes or proteins (Võsa et al. 2021; Sun et al. 2023; Freimann et al. 2025), reflecting a high degree of coregulation in *trans*-QTL networks (Fig. 1c).

To avoid these pleiotropic effects, many studies focus on *cis*-acting genetic variation to identify the putative causal effect of drug target (typically a gene or protein) perturbation on the outcome of interest. This approach is referred to as *cis*-MR. In *cis*-MR, gene expression or protein abundance in an accessible tissue is typically used as an exposure. However, *cis*-MR can still be subject to 2 types of horizontal pleiotropy. First, if the gene or protein affects the outcome in one cell type or developmental stage but is measured in another one, then this can lead to overdispersion heterogeneity or allelic spread that can bias the MR estimates (Patel et al. 2024; Tambets et al. 2024) (Fig. 1a). Fortunately, several pleiotropy-robust MR methods have been developed to address this (Zhu et al. 2021; Tambets et al. 2024; van der Graaf et al. 2025). Secondly, *cis*-MR can also be subject to coregulation between neighboring genes (Fig. 1b) (Tambets et al. 2024). Despite these limitations, *cis*-MR has been successfully used to identify known causal relationships in multiple benchmarks (Porcu et al. 2019; Zheng et al. 2020; Karim et al. 2023; van der Graaf et al. 2025). However, current transcriptomic datasets are limited in sample size for most cell types and tissues (The GTEx Consortium 2020; Kerimov et al. 2023; Tambets et al. 2024), and plasma proteomic studies with large sample sizes only cover a subset of the proteome (eg 2,923 proteins in the UK Biobank (Sun et al. 2023)).

Importantly, the variant effect on gene or protein function can also be captured by its effect on proximal downstream phenotypes in metabolic pathways or regulatory networks (Fig. 1d). For example, for well-known lipid loci, recent *cis*-MR studies have used the variant effect on plasma LDL cholesterol level as a proxy measure for the variant effect on protein function (Lotta et al. 2016; Schmidt et al. 2020; Richardson et al. 2022; Yang et al. 2024). *Cis*-MR, where the exposure is a metabolite or another biomarker, is sometimes also referred to as drug target MR (Schmidt et al. 2020; Gill et al. 2021; Richardson et al. 2022). Similarly, we have used downstream effects of genetic variation on gene expression (*trans*-eQTLs) to characterize the impact of a lupus-associated *USP18* missense variant on its protein function (Freimann et al. 2025). However, a systematic analysis of when and how these proxy measures for gene or protein function can be used for causal inference is still lacking, especially outside of the lipids–cardiovascular domain (Schmidt et al. 2020)

In this study, we expand on the use of high-throughput plasma metabolite measurements as proxy measures for protein function in the *cis*-MR framework. Using genotype and nuclear magnetic resonance (NMR) spectroscopy data from 246,683 UK Biobank participants, we identify 107 confidently fine-mapped missense variants for 56 metabolites. In 2 case studies involving glycolysis and vitamin D synthesis pathways, we demonstrate how the missense variants' effects on pyruvate and histidine level can be used as proxy readouts for their effect on *cis* protein function, allowing us to infer causal relationships between disruption of protein function and downstream traits. Finally, we propose a theoretical framework that outlines the key assumptions that need to be satisfied to generalize this approach to other proteins and traits.

## Methods

### Study cohort

The UK Biobank is a longitudinal biomedical study of approximately half a million participants between 38 and 71 years old from the United Kingdom (Bycroft et al. 2018). Participant recruitment was conducted on a volunteer basis and took place between 2006 and 2010. Initial data were collected in 22 different assessment centers throughout Scotland, England, and Wales. Data collection includes elaborate genotype, environmental, and lifestyle data. Blood samples were drawn at baseline for all participants, with an average of 4 h since the last meal, ie generally nonfasting. NMR metabolomic biomarkers (Nightingale Health, quantification library 2020) were measured from EDTA plasma samples (aliquot 3) during 2019 to 2024 from the entire cohort. Details on the NMR metabolomic measurements in UK Biobank have been described previously for the first tranche of ~120,000 samples (Julkunen et al. 2023). The UK Biobank study was approved by the North West Multi-Centre Research Ethics Committee. This research was conducted using the UK Biobank Resource under application numbers 91233 and 30418.

### Metabolite measurements

This dataset encompassed both the tranche 1 dataset, comprising approximately 130,000 samples, and the tranche 2 dataset, which augmented the resources with an additional 170,000 samples. Details of Nightingale's NMR metabolomics platform and the biomarker measures have been provided for UK Biobank's metabolomics Supplier Criteria Tables in July 2016 (project reference 15004). For the current research, 56 biomarkers from the available panel were selected for genome-wide association study (GWAS) analysis and fine-mapping (Supplementary Table 1). We excluded individuals with more than 5 missing metabolite measurements from the cohort and applied a metabolite-wise inverse normal transformation to obtain the final dataset.

### Principal component analysis-based genetic ancestry assignment

We performed principal component analysis (PCA) of the genotype data using FlashPCA2 (Abraham et al. 2017). Subsequently, all individuals within the UK Biobank dataset who also had NMR data available were clustered into genetic ancestry groups based on their first 3 principal components (PCs) using the GaussianMixture() function from the scikit-learn Python module. The number of mixture components was set to 4 based on empirical analysis. The final dataset, representing the largest PCA cluster corresponding to predominantly European genetic ancestry individuals, comprised 246,683 individuals.

### Association testing and fine-mapping

The association testing between genetic variants and 56 metabolites was conducted using the regenie software (Mbatchou et al. 2021). During the analysis, sex and the 10 top genotype PCs calculated with FlashPCA2 were utilized as study-specific covariates. In regenie step 1, the LD-pruned variants were used as an input. LD pruning was performed with PLINK2 with the following parameters: minor allele frequency (MAF) > 0.001, window size = 50,000 variants, window shift at the end of each step = 200

variants, and pairwise $r^2$ threshold = 0.05. In regenie step 2, the minimum imputation info score was set to 0.6, and the minimum minor allele count was calculated based on the number of samples so that MAF would be equal to 0.001.

After association testing, the statistical fine-mapping on the summary statistics obtained from regenie and in-sample LD matrix was conducted using the Sum of Single Effects (SuSiE) model (Wang et al. 2020). LD matrices were calculated with LDstore2 (Benner et al. 2017) software for each fine-mapped region. Fine-mapped regions were defined for each genome-wide significant locus ($P < 5 \times 10^{-8}$) by considering a 3-Mb wide window centered around the lead variant. In cases where these regions overlapped but did not exceed a total span of 6 Mb, they were merged into a single region. If the resulting region exceeded this 6-Mb limit, the originally defined regions were recursively reduced until all regions adhered to this size constraint. (If LDstore2 encountered a segmentation fault in the following step, alternative maximum region limit of 4.5 or 3 Mb was employed instead.) Regions containing fewer than 50 variants were omitted from the analysis. Additionally, due to the extensive LD structure in the region, the major histocompatibility complex (MHC) region (chr6:28,477,797-33,448,354) was excluded from fine-mapping. In the SuSiE method (Wang et al. 2020), the maximum number of causal variants within a locus was set to 10. Consequently, up to 10 independent 95% credible sets (CSs) and posterior inclusion probabilities (PIPs) for each variant were computed, utilizing the default uniform prior probability of causality.

Association testing and fine-mapping were performed on the human genome assembly GRCh37. Subsequently, the coordinates of the imputed variants within the fine-mapping results were lifted to the GRCh38 build. This transition was accomplished using the liftover() function available in the R package MungeSumstats (Murphy et al. 2021). The Nextflow workflow for GWAS analysis and fine-mapping is available on GitHub (https://github.com/AlasooLab/reGSusie).

### Mendelian randomization

For the primary *cis*-MR analysis, we used the fine-mapped missense variants as instruments and used fitSlope() function from the MRLocus R package version 0.0.26 to perform inference (Zhu et al. 2021). To explore the impact of instrument selection on *cis*-MR analysis, we further performed greedy LD pruning around the *PKLR* and *HAL* genes. For this analysis, we included only variants from the +/−200 kb region around the gene body of the target gene that had MAF > 1%. For instrument selection, we used a greedy pruning strategy to only retain variants with $P < 5 \times 10^{-8}$ and pairwise $r^2 < 0.01$. With the greedily pruned instruments, we performed *cis*-MR analysis using both multiplicative random-effects inverse variance weighted MR (IVW-MR) implemented in the MendelianRandomization R package (Yavorska and Burgess 2017) and MRLocus. For the IVW-MR method, we also specified "weights = delta."

### Results

We performed a GWAS and fine-mapping for 56 metabolites in the UK Biobank using the NMR platform from Nightingale Health (see Methods). The analysis included 246,683 individuals of European ancestries (see Methods). In total, we identified 107 confidently fine-mapped (PIP > 0.8) missense variants that were associated with 1 or more metabolites. All summary statistics and fine-mapping results are publicly available (see Data availability). Below, we will present 2 case studies: one focusing on the effect of glycolysis pathway activity on red blood cell count and another one exploring the role of histidine ammonia lyase (*HAL*) gene in modulating vitamin D level.

### Glycolysis pathway, plasma pyruvate level, and red blood cell count

Loss-of-function mutations in the pyruvate kinase L/R (*PKLR*) gene are the most common cause of hemolytic anemia, a disorder in which red blood cells are destroyed faster than they are made (Zanella et al. 2007). In our analysis, we identified 8 missense variants (including 2 variants in the *PKLR* gene) that were robustly associated with plasma pyruvate level (Fig. 2a). Reassuringly, the 2 *PKLR* variants (1_155291845_C_T, rs113403872, and 1_155291918_G_A, rs116100695) were also associated with red blood cell (RBC) count, thus confirming the known disease association (Zanella et al. 2007). The 2 *PKLR* variants were not in high linkage disequilibrium (LD) with each other (Supplementary Table 2). Despite the strong association at the *PKLR* locus, there is no obvious causal mechanism directly linking the level of circulating pyruvate to RBC count. However, when performing MR between plasma pyruvate level and RBC count using all 8 missense variants as instruments, we detected a nonzero "causal" effect (Fig. 2b). Notably, there was considerable heterogeneity among the causal effect estimates (Wald ratio) provided by individual genetic instruments, prompting further investigation.

We noticed that in addition to the 2 *PKLR* missense variants, 2 more missense variants affected another core enzyme of the glycolysis pathway (12_48118502_C_A, rs4760682 in *PFKM* and 21_44326728_C_T, rs118106526 in *PFKL*, both encoding the phosphofructokinase enzyme) (Fig. 2c; Supplementary Fig. 1). Given that mature RBCs lack both nuclei and mitochondria, their energy production, which is essential for their survival, relies entirely on the glycolysis pathway (van Wijk and van Solinge 2005). As the end product of the glycolysis pathway is pyruvate, we hypothesized that for these 4 missense variants, plasma pyruvate level might serve as a proxy readout for the glycolysis pathway activity in RBCs (see causal diagram on Fig. 2d).

Indeed, for the 4 missense variants in the *PFKM*, *PFKL*, and *PKLR* genes, we observed directionally concordant effects between reduced plasma pyruvate level and decreased RBC count (Fig. 2a). This was further supported by MR, which now showed considerably better concordance between the causal effect size (Wald ratio) estimates provided by the individual variants (Fig. 2d; Supplementary Fig. 2). Finally, we observed very similar causal effect estimates when using greedy LD pruning to select instruments only around the *PKLR* gene (Supplementary Fig. 3). Importantly, we were now seeking to infer the effect of glycolysis pathway activity on RBC count, rather than the effect of circulating pyruvate level. Hence, we are using the variants' effects on plasma pyruvate level only as a proxy to capture their effects on glycolysis pathway activity in RBCs.

As a final validation, we repeated the MR analysis using the 4 missense variants in genes that do not encode enzymes directly involved in the glycolysis pathway (*GCKR*, *NDOR1*, *AMPD3*, *PDK3*) and detected a null effect (Supplementary Fig. 4), indicating that the initial genome-wide MR estimate (Fig. 2b) was primarily driven by the missense variants in genes encoding enzymes of the glycolysis pathway. Notably, the glucokinase regulator (*GCKR*) missense variant (rs1260326, *GCKR*:p.Leu446Pro) is a highly pleiotropic locus associated with 51 (out of 56) selected metabolites in our recent meta-analysis of 599,249 individuals (Tambets et al. 2025). This example highlights how the levels of plasma metabolites can be regulated through multiple distinct

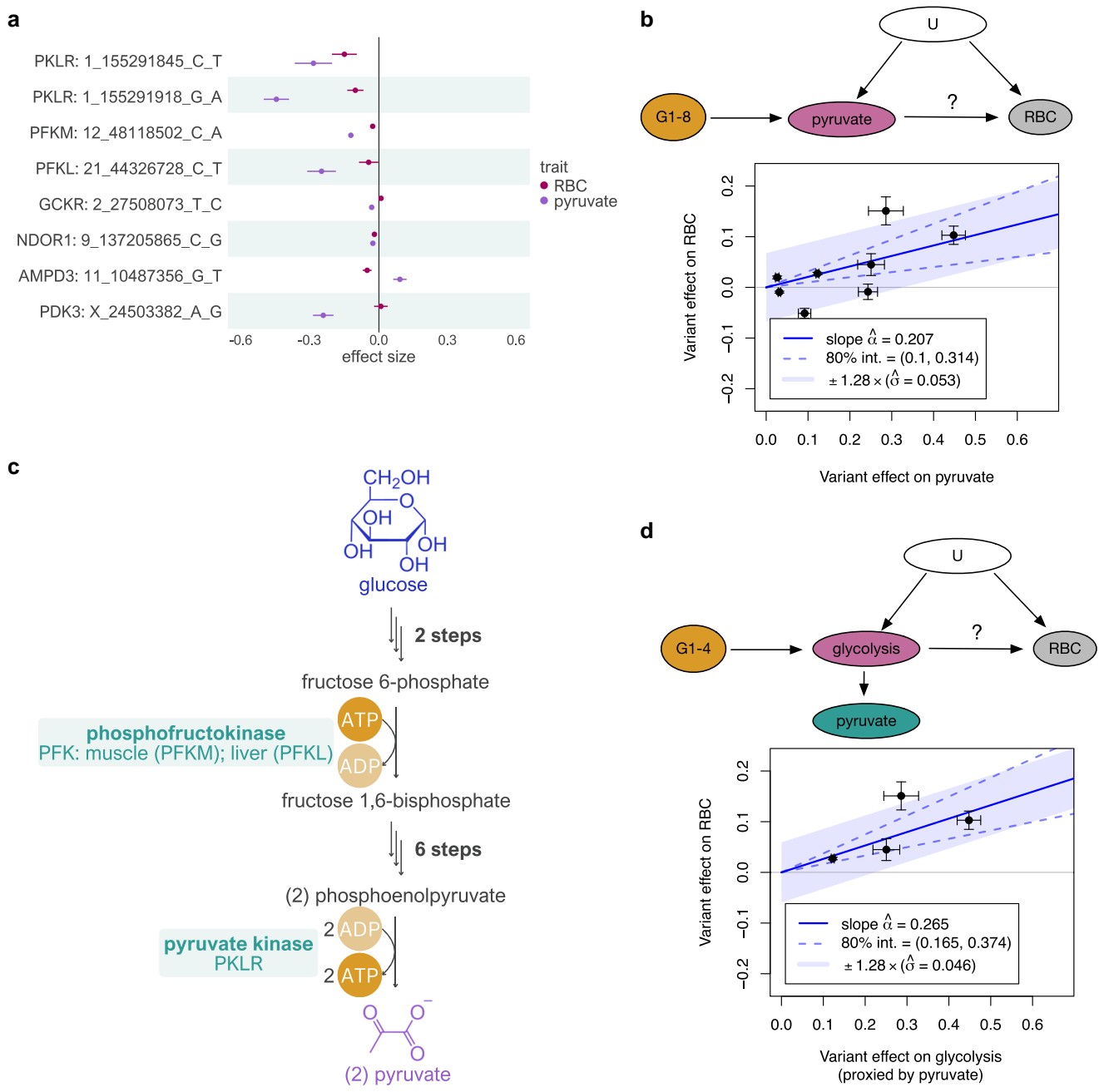

**Fig. 2.** Relationship between plasma pyruvate level and RBC count. a) Eight fine-mapped missense variants associated with plasma pyruvate level and their effect on RBC count. Pairwise LD for the *PKLR* missense variants is shown in Supplementary Table 2 b) MR between plasma pyruvate level (exposure) and RBC count (outcome) using all 8 fine-mapped missense variants as instruments. c) Role of phosphofructokinase (encoded by *PFKM* and *PFKL* genes) and pyruvate kinase (encoded by *PKLR*) in the glycolysis pathway. Complete pathway is shown in Supplementary Fig. 1. d) MR between glycolysis pathway activity (exposure) and RBC count (outcome), restricted to missense variants in the 3 genes (*PKLR*, *PFKM*, and *PFKL*) that encode enzymes involved in the glycolysis pathway. The causal diagram illustrates how plasma pyruvate level acts as a proxy for glycolysis pathway activity in RBCs. G, genetic instruments; U, unmeasured confounders.

mechanisms. However, even if the metabolite itself (eg pyruvate) is unlikely to have a direct causal effect on the outcome of interest (RBC count), it can still act in a locus-specific manner as a proxy measure for other biological traits (eg glycolysis) that do have a causal effect.

## Histidine, UV exposure, and vitamin D

We recently noticed an interesting common variant (MAF = 42%) GWAS hit near the *HAL* gene (12_95984993_C_T, rs3819817) that was associated both with vitamin D level (Manousaki et al. 2020) and skin cancer (Seviiri et al. 2022). In the Open Targets Genetics portal (Mountjoy et al. 2021), this variant was identified as an eQTL for the *HAL* gene and was also associated with *trans*-urocanate level in urine (Schlosser et al. 2020) and childhood sunburn occasions (Neale Lab), but was not pleiotropically associated with any other disease (Fig. 3a, and b). Furthermore, the lead variant had a positive effect on *HAL* expression and *trans*-urocanate level and a negative effect on vitamin D level, sunburn occurrences, and skin cancer risk (Fig. 3c).

The biochemical role of the HAL protein in regulating vitamin D level is well understood (Fig. 4). HAL is an enzyme that converts histidine to *trans*-urocanate (Hall 1952). As a natural sunscreen, *trans*-urocanate absorbs UV light and isomerizes to its *cis*-form. This process reduces the effective UV radiation dose in humans, thereby inhibiting vitamin D synthesis. However, the lower dose may also provide protection against sunburn and skin cancer (Barresi et al. 2011). In the liver, *trans*-urocanate is further converted by the urocanate hydratase (encoded by *UROC1*) into 4-imidiazolone-5-propionate (Fig. 4) (Kessler et al. 2004). Interestingly, this conversion does not occur in the skin, as *UROC1* is highly expressed in the liver (median transcripts per

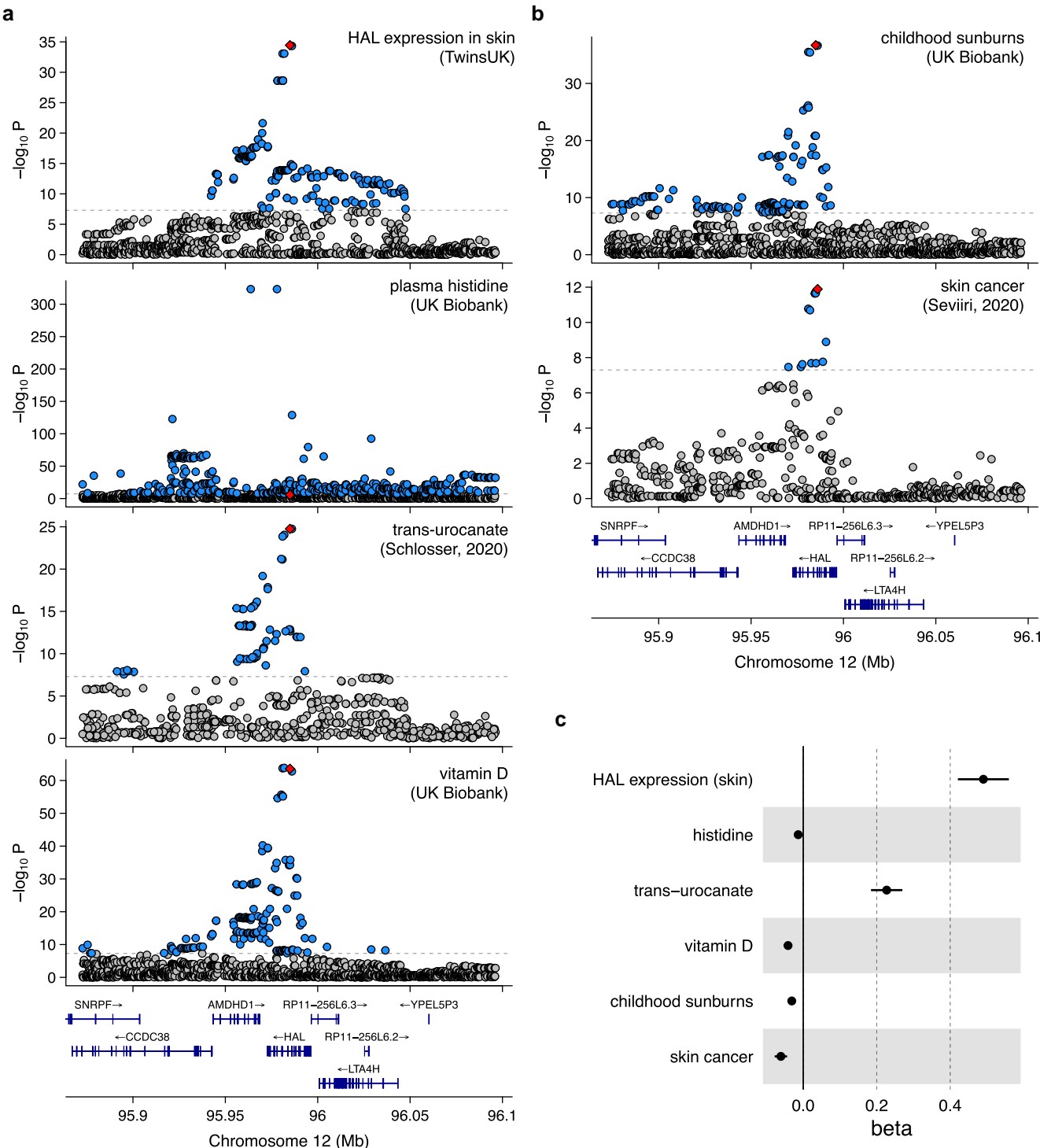

**Fig. 3.** Effect of skin-specific regulatory variation on vitamin D level and other related traits. a) Regional association plots for *HAL* expression in skin, plasma histidine, *trans*-urocanate, and vitamin D levels, illustrating statistically significant associations within the same genomic region. The lead *HAL* eQTL variant (rs3819817) has been highlighted. b) Regional association plots for childhood sunburns and skin cancer in the same genomic region. c) Effect size of the rs3819817 *HAL* eQTL lead variant on the 6 traits.

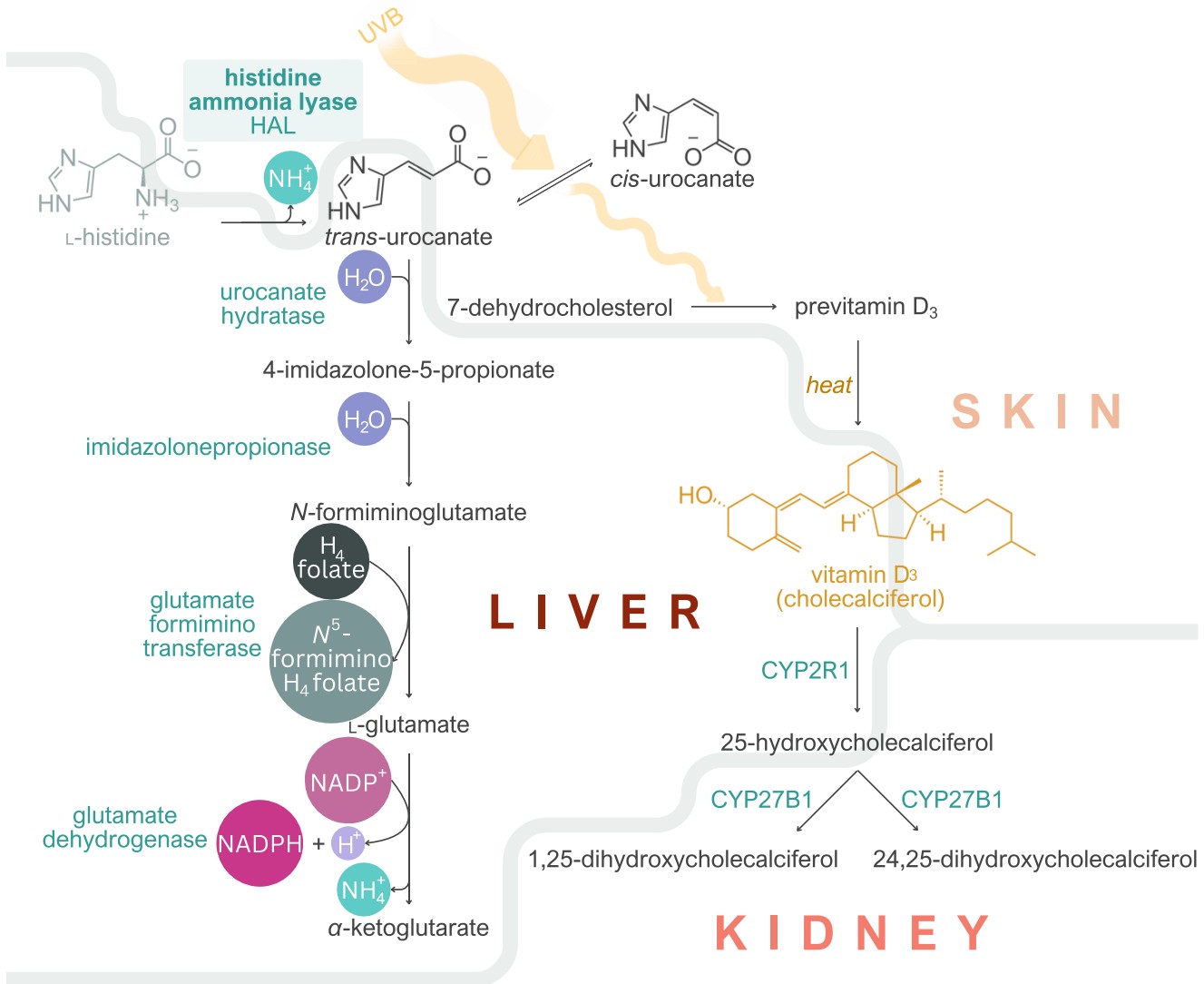

**Fig. 4.** The role of histidine metabolism in regulating vitamin D level. Role of HAL in regulating vitamin D level in a skin-specific manner.

million (TPM) = 75.6 in GTEx) but not in the skin (median TPM = 0.01) (The GTEx Consortium 2020) leading to *trans*-urocanate accumulation in the skin and thus to reduced vitamin D level but also protection from skin cancer (Fig. 3c).

Unexpectedly, although we anticipated that the effect of the rs3819817 *HAL* eQTL variant on vitamin D level and skin cancer risk would be mediated by the conversion of histidine to *trans*-urocanate (Fig. 4), the variant had only a weak association with plasma histidine level (beta = −0.014; $P = 1.8 \times 10^{-6}$; Fig. 3c). To understand this discrepancy, we examined the rs3819817 *HAL* eQTL effect sizes and *P*-values in all 127 datasets in eQTL Catalogue release 6 (Kerimov et al. 2023). The eQTL was highly tissue specific and detected only in 3 skin datasets from the TwinsUK (Buil et al. 2015) and GTEx (The GTEx Consortium 2020) studies (Supplementary Fig. 5). This suggests that the rs3819817 *HAL* eQTL variant primarily affects histidine level in the skin rather than in plasma, via tissue-specific regulation of *HAL* gene expression.

Since histidine was one of the 56 metabolites profiled in our analysis, we next focused on fine-mapped missense variants associated with plasma histidine. Reassuringly, 2 of the strongest associations corresponded to 2 low-frequency (MAF < 0.5%) missense variants in the *HAL* gene (12_95977953_C_T, rs61937878, and 12_95986106_C_T, rs117991621), which were also strongly associated with vitamin D level (Fig. 5a) (Kanai et al. 2021). We hypothesized that for the 2 *HAL* missense variants, we could use their effect on reducing plasma histidine level as a proxy measure for their effect on HAL protein function. Using this approach with MR, we detected a significant causal effect of −0.152 between increased HAL protein function (proxied by reduction in plasma histidine) and vitamin D level (Fig. 5b). This estimate remained unchanged when histidine was measured in the Estonian Biobank (Tambets et al. 2025) instead of the UK Biobank (beta = −0.163; Supplementary Fig. 2). Furthermore, using GWAS summary statistics from a skin cancer meta-analysis across the FinnGen, Million Veterans Program, and UK Biobank (FinnGen + MVP + UKBB) biobanks (MVP–FinnGen–UKBB Meta-Analysis 2024) as an outcome, we also detected a proportional effect of these 2 missense variants on reducing skin cancer risk (beta = −0.192; Supplementary Fig. 6), mirroring the common *HAL* eQTL variant associations discussed above (Fig. 3c).

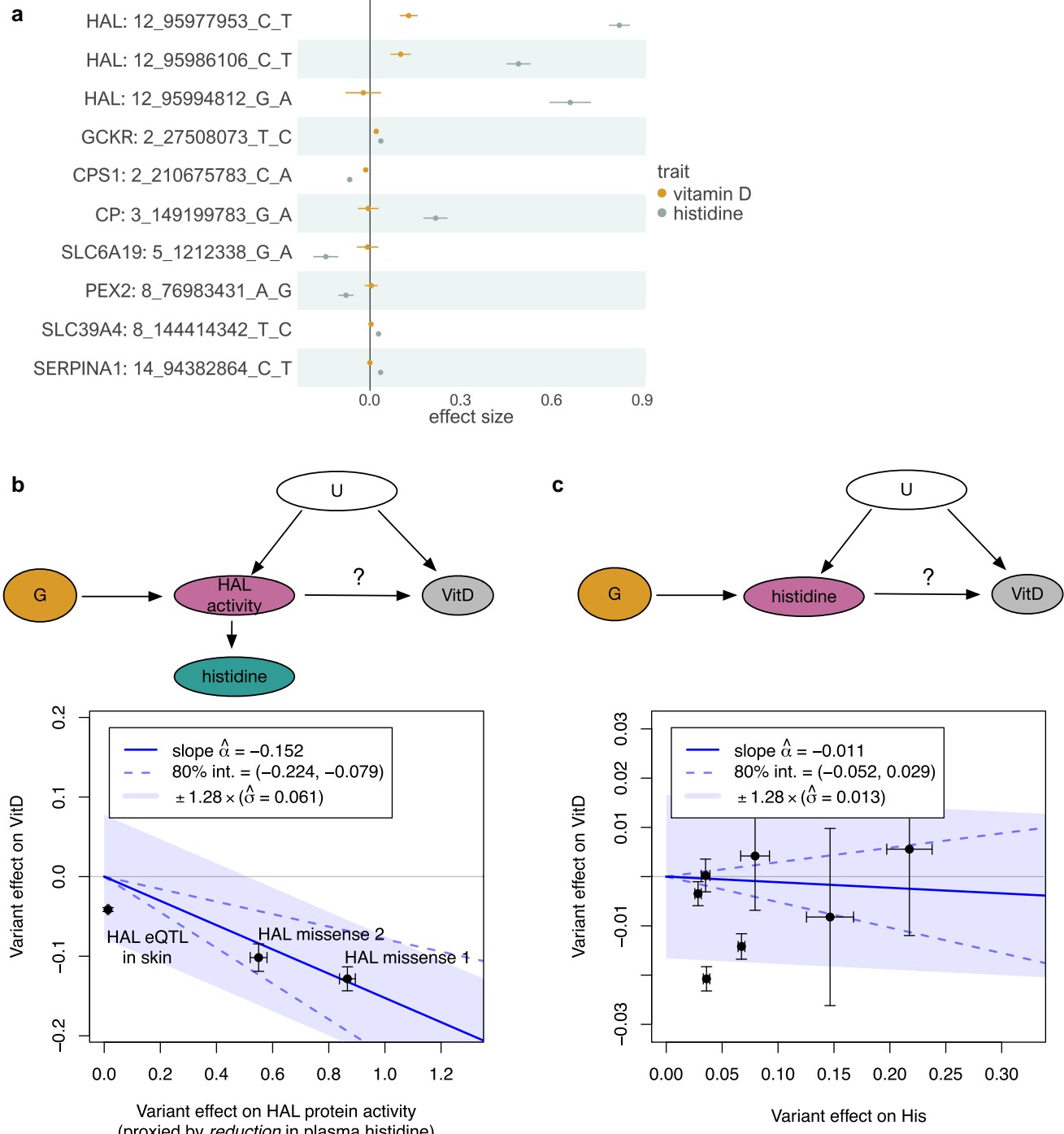

**Fig. 5.** Using plasma histidine level as a proxy for HAL protein activity. a) Fine-mapped (PIP > 0.8) missense variants associated with plasma histidine level in the UK Biobank. Pairwise LD for the *HAL* missense variants is shown in Supplementary Table 2. b) MR analysis examining the relationship between proxied HAL protein activity and vitamin D level. The instruments are restricted to the 2 missense variants in the *HAL* gene and a skin-specific eQTL for *HAL* (Fig. 3a). Here, we use the effect of these variants on reducing plasma histidine level as a proxy measure for their effect on HAL function. c) MR between plasma histidine and vitamin D level using all fine-mapped missense variants associated with plasma histidine level outside the *HAL* region as instruments. G, genetic instruments; U, unmeasured confounders.

Curiously, our fine-mapping also identified a third missense variant (12_95994812_G_A, rs143854097) in the *HAL* gene with an even lower allele frequency (MAF ~ 0.1%) that was only associated with histidine level and not with vitamin D level (Fig. 5a). This variant was not imputed in the Estonian Biobank, so we could not replicate its effect on plasma histidine in an independent

sample. The variant was also not included in the FinnGen + MVP + UKBB meta-analysis for skin cancer, so we could also not assess its effect on skin cancer risk. Given that the rs143854097 variant had a discordant effect relative to the other 2 missense variants as well as the *HAL* eQTL variant, we suspect that the lack of association with vitamin D might

represent either a technical artifact due to its low allele frequency or an unknown tissue-specific mechanism. For completeness, including the rs143854097 missense variant into the vitamin D MR analysis attenuated the effect size estimate (Supplementary Fig. 7).

An important feature of our MR analysis is that plasma histidine only acts as a proxy readout of variant effect on HAL function (see diagram on Fig. 5a) and is unlikely to have a direct causal effect on vitamin D. This raises the question of which histidine-associated variants are valid instruments for causal inference. Most obviously, variants outside the *HAL* locus should not exhibit consistent effects on vitamin D, which was indeed the case (Fig. 5, a and c). However, things are also more complicated within the *HAL* locus. For example, using the skin-specific eQTL variant (rs3819817) with plasma histidine level as exposure would have yielded a highly misleading estimate of −0.0414/0.014 = −2.96 (Wald ratio) (Fig. 5b). This is because this variant likely has a much larger effect on HAL function in the skin, the causal tissue for vitamin D level, than in the tissues that determine histidine level in plasma. Interestingly, using the expression level of *HAL* in the skin as the exposure (instead of plasma histidine level) with the same instrument yielded a causal effect estimate of −0.0414/0.49 = −0.084 (Wald ratio), which aligns more closely with the estimate from the 2 missense variants (−0.152). Alternatively, using greedy LD pruning to select instruments around the *HAL* gene (with plasma histidine as exposure) also yielded a null MR estimate (Supplementary Fig. 8), suggesting that regulatory variants could often have discordant effects on HAL function in the skin compared to other tissues that determine plasma histidine level. Thus, the need to consider tissue- and cell-type-specific effects poses a significant limitation to using variant effects on circulating metabolites (or other molecular traits) as proxy measures for protein function.

## Additional assumptions of MR with proxy exposures

Both the glycolysis and vitamin D examples illustrate how restricting genetic instruments to specific gene regions and using variant effects on plasma metabolites as proxy measures for corresponding gene function can help reduce horizontal pleiotropy and infer plausible causal relationships between perturbed gene function and outcomes of interest. However, generalizing this approach to other gene regions and potential proxy exposures requires careful consideration of 2 key assumptions:

1. **The instruments (genetic variants) must be unambiguously linked to the causal *cis*-gene**. Most trait-associated genetic variation is noncoding, likely modulating the expression or splicing of nearby *cis*-genes. We and others have shown that expression-altering variants often regulate the expression of multiple neighboring genes (Tambets et al. 2024) (Fig. 1b). Although splicing QTLs tend to have more specific effects on a single target gene, distinguishing them from expression QTLs can be challenging in practice (Kerimov et al. 2023). This is the main reason why we focused on fine-mapped missense variants in this study, as they can be linked to the causal gene with high confidence. However, missense variants are rare and may not be available for most traits and exposures. Thus, potential violation of this assumption should be explicitly considered when performing analyses such as drug target MR that include all genetic variants from a specific gene region as instruments (Richardson et al. 2022; Gill et al. 2024; Yang et al. 2024).

2. **For accurate inference, the proxy metabolite, transcript, or protein being measured should be *downstream* and *proximal* to the *cis*-gene or protein of interest whose function we are aiming to approximate**. In our case study, for variants affecting the *HAL* gene, it is preferable to use histidine or *trans*-urocanate concentrations rather than metabolites further downstream in the pathway (Fig. 4). In practice, however, the exact mechanisms by which the *cis*-gene affects the measured traits are often unclear, which could inadvertently result in capturing traits that are downstream of the outcome of interest, potentially leading to reverse causation. As GWAS sample sizes increase, the proportion of discoveries that correspond to these indirect effects is also likely to increase. For example, in a very large meta-analysis of NMR metabolites ($n = 599,249$), the *HAL* missense variant rs61937878 was also weakly associated with plasma glycine level (beta = 0.071; $P = 2.5 \times 10^{-10}$), likely reflecting an indirect pleiotropic effect (Supplementary Fig. 9).

In addition to these 2 assumptions specific to proxy exposures, we also need to consider the factors that can invalidate any *cis*-MR analysis with molecular traits as exposures. First, molecular traits such as gene expression, protein abundance, or metabolite concentrations can often be measured in many different cell types, tissues, or developmental stages (contexts for short). In an ideal scenario, the context in which the genetic variant's effect on the exposure has a causal effect on the outcome ("causal context") is the same where the exposure is measured ("proxy context"), but this is often not the case. In the *HAL* example, the likely causal context where *HAL* influences vitamin D level is skin tissue, but the proxy context in which histidine was measured is plasma. If a genetic variant has the same effect on the exposure in the proxy context as it would in the causal context (eg it is a missense variant), then context misspecification is less important. However, noncoding regulatory variants can often have context-specific effects, and this can significantly bias MR estimates. For example, the skin-specific eQTL for *HAL* had almost no effect on plasma histidine level (Fig. 3a). Conversely, using greedy LD pruning to blindly select genetic instruments around the *HAL* gene missed the effect of HAL protein function on vitamin D, probably due to tissue-specific effects (Supplementary Fig. 8). Secondly, even if the included instruments themselves do not have context-specific effects, they might still be in LD with other context-specific genetic variants that do. This can bias the marginal effect sizes of the instruments on the exposure, thus also biasing the MR estimates when using an exposure measured in the proxy context. A promising approach to account for these biases is using methods such as MR-link-2 that explicitly model the LD between instruments and their potentially pleiotropic effects (van der Graaf et al. 2025).

## Guidelines for performing *cis*-MR with proxy exposures

To make it easier to perform *cis*-MR studies with proxy exposures and avoid some of the common pitfalls, we have put together the following guidelines also illustrated on Fig. 6:

**Step 1.** Familiarize yourself with the best practices and pitfalls of *cis*-MR and drug target MR. Some excellent places to start are (Burgess et al. 2019; Schmidt et al. 2020; Gill et al. 2021; Sanderson et al. 2022; Gill et al. 2024) and references contained therein.

**Step 2**. Assess whether the target protein has a plausible proximal, downstream readout that reflects its biological function.

**Step 3.** Assess if the proxy exposure is likely to be on the causal pathway from the protein to the outcome of interest (eg LDL for

## Guidelines for performing *cis*-MR analysis with proxy exposures

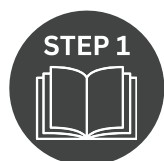

**STEP 1** Familiarise yourself with the best practices and pitfalls of *cis*-MR and drug target MR.

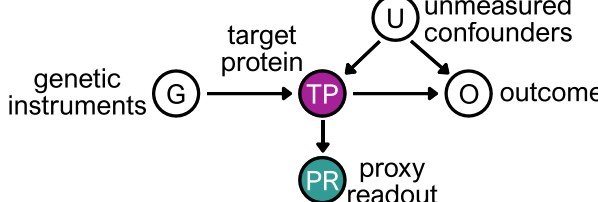

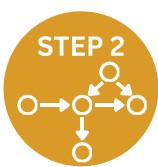

**STEP 2** Assess whether the target protein has a plausible proximal, downstream readout that reflects its biological function. **If no such readout exists, or if the genetic instruments are not strongly associated with it, *cis*-MR should not be pursued.**

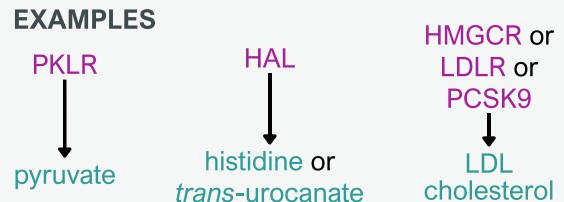

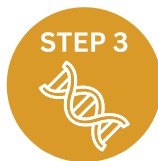

**STEP 3** To select the genetic instruments, determine whether the proxy exposure **(A)** is on the causal pathway from the protein to the outcome (vertical pleiotropy), or **(B)** is measured in a non-causal context.

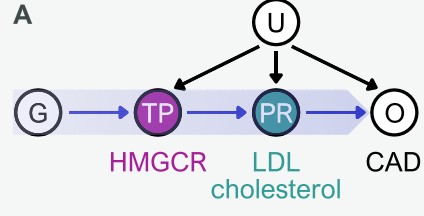

*CAD – coronary artery disease*

Relying on simple LD pruning for variant selection is likely to be sufficient.

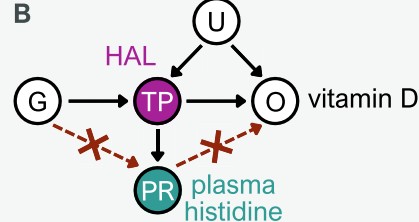

Restrict genetic instruments to variants that have consistent effects on target protein function across cell types and contexts, such as fine-mapped missense or loss-of-function variants.

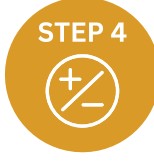

**STEP 4** Use positive and negative control outcomes to validate the genetic instruments.

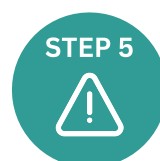

**STEP 5** Recognize that nearly all protein function readouts (e.g., gene or protein abundance) are proxies and may not fully capture true function due to cell-type mismatch or weak correlation.

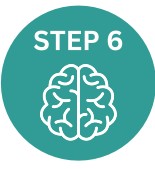

**STEP 6** Understand that no checklist guarantees robust causal inference. Reliable conclusions require critical evaluation and triangulation of evidence from multiple sources. **Importantly, know when MR is not appropriate despite available data.**

**Fig. 6.** Guidelines for performing *cis*-MR analysis with proxy exposures.

coronary artery disease) or if it is measured in a noncausal context (eg plasma histidine for vitamin D). As highlighted by our *HAL* case study, proxy exposures measured in noncausal contexts might require restricting instrument selection to variants that have the same effect on protein function in all contexts (eg missense and putative loss-of-function variants).

**Step 4**. Use positive and negative control outcomes to validate genetic instruments (Sanderson et al. 2021; Jiesisibieke et al. 2025). Selecting the most appropriate controls for *cis*-MR will likely require extensive domain knowledge.

**Step 5**. Recognize that nearly all protein function readouts (eg gene or protein abundance) are proxies and may not

capture true function due to technical artifacts or cell-type mismatch.

**Step 6.** Understand that no checklist can guarantee robust causal inference. Sometimes the most important insight might be realizing when MR should not be performed (Jiesisibieke et al. 2025; Reed et al. 2025).

## Discussion

Using examples from the glycolysis and vitamin D synthesis pathways, we have constructed 2 case studies to demonstrate how horizontal pleiotropy can mislead MR to infer implausible causal relationships between an exposure and an outcome. Our case studies complement previous reports highlighting widespread horizontal pleiotropy affecting plasma metabolite levels and other high-throughput molecular measurements (Richardson et al. 2022; Smith et al. 2022; Karjalainen et al. 2024; Yang et al. 2024; Freimann et al. 2025). We illustrate how MR analysis can be reformulated by focusing on genetic variation located in *cis* of specific target genes and using the high-throughput molecular measurements as proxy readouts of protein function (Fig. 1e). The key contribution of our work is to explicitly outline the additional assumptions required for this approach to produce valid inferences. We expand on previous work focused on well-known lipid loci (Richardson et al. 2022; Yang et al. 2024) by providing a general framework for conducting MR analysis using arbitrary proxy measures of protein function.

A related "*trans*-weighted *cis*-MR" idea was presented in the MR-Fish study (Warwick et al. 2024). However, a key difference between our analysis and theirs is that they did not explicitly consider the assumptions that the instruments and proxy exposures should satisfy to produce reliable inferences. For instance, by using variants in the *FTO* locus as instruments and plasma CRP level as the proxy exposure, the authors inferred a putative causal link between altered *FTO* function (proxied by variant effect on CRP) and type 2 diabetes risk. However, they overlooked that the lead noncoding variant at the *FTO* locus regulates the expression of *IRX3* and *IRX5* transcription factors instead of the *FTO* gene itself (Claussnitzer et al. 2015), thereby violating our first assumption. Thus, it is unclear what is the added value of the MR-Fish approach beyond simply reporting the closest genes at the outcome-associated locus, because there is no guarantee that the included instruments have any effect on the claimed *cis*-gene. They also did not consider our "proximal effect" assumption (assumption 2), which could easily lead to cases of reverse causation, where the variant effect on the exposure is mediated via the outcome.

Most *cis*-MR analyses use gene expression levels or protein abundances as exposures (Porcu et al. 2019; van der Graaf et al. 2020; Zheng et al. 2020; Karim et al. 2023; Tambets et al. 2024). However, when the aim is to infer the causal relationship between altered protein function and an outcome of interest, gene expression and protein abundance are themselves imperfect proxies of protein function. This could be particularly problematic for missense and splice regulatory variants, as their effect on gene expression and protein abundance might be poorly correlated with protein function due to the existence of distinct functional isoforms (Park et al. 2018; Wright et al. 2022; Gotthardt et al. 2023) or because of assay-specific quantification artefacts (Pietzner et al. 2021; Eldjarn et al. 2023). Using downstream regulatory or metabolic effects as proxy measures for protein function mitigates these limitations. For example, using *HERC5* gene expression in lymphoblastoid cell lines as a readout of the missense variant effect on USP18 protein function allowed us to establish a potentially causal link between reduced USP18 function and increased lupus risk (Freimann et al. 2025). This would not have been feasible using the standard *cis*-MR approach, as the missense variant had no effect on *USP18* gene expression (there was no *cis*-eQTL), and USP18 protein abundance has not been measured in the disease-relevant context.

Using proxy exposures in *cis*-MR also has several limitations, as outlined by the assumptions above. In particular, the variant mechanisms of action for most detected genetic signals are often unknown, making it challenging to unambiguously link the variants to the causal genes. Furthermore, for most metabolite GWAS signals and *trans*-QTL loci, we lack sufficient mechanistic understanding to determine whether the detected effect is proximal or indirectly mediated by other factors. In fact, in our 2 case studies, our focus on using pyruvate and histidine as proxy exposures was largely guided by the names of the 2 relevant enzymes, *pyruvate* kinase and *histidine* ammonia lyase, which pointed to plausible links. These limitations can restrict the practical utility of using proxy exposures in MR, and we caution against performing automated all-against-all *cis*-MR analyses with proxy exposures without careful consideration of the underlying assumptions.

## Data availability

The GWAS summary statistics for the 56 metabolic traits are available from Zenodo (https://doi.org/10.5281/zenodo.13821209). The fine-mapped credible sets and log Bayes factors from SuSiE are available from Zenodo (https://doi.org/10.5281/zenodo.13821038).

The UK Biobank summary statistics and fine-mapping results for RBC count and vitamin D (Kanai et al. 2021 ) are available from Zenodo (https://doi.org/10.5281/zenodo.17182011). The GWAS summary statistics for skin cancer from Seviiri et al. (2022) were downloaded from the GWAS Catalog (accession GCST90137411). The skin cancer summary statistics from the FinnGen–MVP–UKBB meta-analysis (phenotype C3_OTHER_SKIN_EXALLC) were downloaded from the FinnGen website (https://mvp-ukbb.finngen.fi/about). The GWAS summary statistics for "childhood sunburn occasions" (UK Biobank data field 1737) were downloaded from the Neale lab website (https://github.com/Nealelab/UK_Biobank_GWAS). The histidine and pyruvate GWAS summary statistics from the Estonian Biobank were downloaded from the GWAS Catalog (accessions GCST90449390 and GCST90449502).

Data analysis code is available from GitHub (https://github.com/AlasooLab/MR_with_proxy_exposures). The GWAS and fine-mapping Nextflow workflow is available from GitHub (https://github.com/AlasooLab/reGSusie). All forest plots were made with the ggforestplot R package (https://github.com/NightingaleHealth/ggforestplot).

Supplemental material available at GENETICS online.

## Acknowledgments

We thank Adriaan van der Graaf and Zoltan Kutalik for their helpful comments on the manuscript. Nightingale Health Plc is acknowledged for early access to the UK Biobank NMR metabolite data. This research has been conducted using the UK Biobank Resource under application numbers 91233 and 30418.

## Funding

IR was supported by the Estonian Research Council (grant no. PSG415). KA and RT were supported by the Estonian Research Council (grant nos. PSG415 and MOB3ERC115).

## Conflicts of interest

E.B.F. is an employee of Pfizer. The remaining authors declare no conflict of interest.

## Author contributions

I.R. performed genome-wide association testing and fine-mapping on the UK Biobank data. R.T. perform colocalization on the summary statistics from the *HAL* locus. E.B.F. initially identified the association at the *HAL* locus and provided a biological interpretation. K.A. conceived the study and performed the MR analyses. I.R. and K.A. wrote the manuscript with contributions from all authors.

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

*Editor: C. Yang*