## [Peer Review File · Genetics]

Mendelian randomisation with proxy exposures: challenges and opportunities

Kaur Alasoo, Ida Rahu, Ralf Tambets, and Eric Fauman

NOTE: The reviews and decision letters are unedited and appear as submitted by the reviewers.

In extremely rare instances and as determined by a Senior Editor or the EIC, portions of a review may be redacted. If a review is signed, the reviewer has agreed to no longer remain anonymous.

The review history appears in chronological order.

Review Timeline:

Submission Date:	2025-05-27
Editorial Decision:	2025-06-22
Resubmission Received:	2025-08-22
Editorial Decision:	2025-09-12
Resubmission Received:	2025-09-13
Accepted:	2025-09-18

June 22, 2025

GENETICS-2025-308224

Mendelian randomisation with proxy exposures: challenges and opportunities

Dear Dr. Alasoo:

Two experts in the field, along with myself, have reviewed your manuscript. We all find the concept of using proxy exposures to be quite intriguing. However, the manuscript is not currently suitable for publication in GENETICS. We would welcome a substantially revised manuscript. Both reviewers have comments and concerns to be addressed in a revised manuscript. You can read their reviews at the end of this email.

In the revision, we hope that you can address the reviewers' comments, in particular, including the following points:

1. Provide clear, practical instructions for choosing genetic instruments and proxy exposures. This is very critical for real applications.
2. Clearly explain how proxy exposures reduce pleiotropy compared to direct genetic measures
3. Explain how this work differs from the similar 2020 study by Schmidt et al. Genetic drug target validation using Mendelian randomisation. Nat Commun 11, 3255 (2020).
4. Explain how they dealt with cell-type-specific effects in case studies.

We look forward to receiving your revised manuscript. Please let the editorial office know approximately how long you expect to need for revisions.

Upon resubmission, please include:

1. A clean version of your manuscript;
2. A marked version of your manuscript in which you highlight significant revisions carried out in response to the major points raised by the editor/reviewers (track changes is acceptable if preferred);
3. A detailed response to the editor's/reviewers' feedback and to the concerns listed above. Please reference line numbers in this response to aid the editor and reviewers.

Your paper will likely be sent back out for review.

Additionally, please ensure that your resubmission is formatted for GENETICS
<https://academic.oup.com/genetics/pages/general-instructions>

Follow this link to submit the revised manuscript: Link Not Available

Sincerely,

Can Yang
Associate Editor
GENETICS

Approved by:
Hongyu Zhao
Senior Editor
GENETICS

Reviewer #1 :

Review Comments

This paper focuses on the problem of horizontal pleiotropy in the cis-MR studies. Review Comments. This paper focuses on the problem of horizontal pleiotropy in the cis-MR studies. The authors present two case studies where the metabolites, as downstream phenotypes of target genes' biological processes, can be used as proxy exposures to infer the causal effects of the genes on outcome traits. This method has two advantages: the sample size of metabolite association studies is usually much larger than the eQTL/pQTL studies; and the metabolite as proxy exposure is less affected by horizontal pleiotropy. The authors also highlight the assumptions needed in such proxy-based cis-MR studies.

Overall, the manuscript is well-structured, provides biological insights into the usage of proxy-based cis-MR, and summarizes

the benefits and limitations of this approach, which is valuable for guiding the MR society in using this class of method. I have several comments regarding the clarity of the analysis process and results.

1. For the MR analysis shown in Figure 5B, which variants were used as instruments to obtain the slope -0.152? Were all 3 missense variants used as instruments? Was the HAL eQTL variant also included as an instrument?
2. How to choose instruments in the proxy-based MR analysis? I understand that using the missense variants is preferred due to their clear relevance to the target gene. In general, if one hopes to apply the proxy-based method, what kind of criteria would be needed to filter the instruments? Can the authors provide a suggested procedure to select valid instruments for proxy exposure?
3. As the authors noted, instruments are usually highly correlated in cis-MR. For example, two missense variants of PKLR gene are used as instruments in the first study and three missense variants of HAL gene are used in the second study. What is the correlation among these variants? Is the LD accounted for in the MR analyses?
4. In the second example, the causal association of interest is the effect of HAL activity on Vitamin D, which happens in skin. However, the proxy exposure, histidine, was measured in plasma. How should one interpret the causal effect when the proxy is measured in a tissue different than the target process?
5. The authors caution the general application of this approach to arbitrary gene regions and proxy exposure pairs. In MR study of a pair of complex traits, one can use established null pairs to benchmark the calibration of a method, e.g., using the hair color as outcome. Is there a parallel approach to calibrate cis-MR, which might help rule out the invalid proxy exposures?
6. Can the estimated causal effects be reproduced when using a different metabolite study?
7. Panels C and D are mislabeled in the caption of Figure1 (line53-54). Should be "(D) the same scenario as in C"

Reviewer #2 :

The authors propose the use of proxy exposures in cis-Mendelian randomization (cis-MR) analysis as a strategy to overcome limitations in traditional MR approaches. This method is illustrated through two case studies focusing on glycolysis and vitamin D synthesis pathways. Several key aspects require further clarification:

- (1) The study suggests that proxy exposures help minimize horizontal pleiotropy, but the rationale behind this claim is not fully developed. For instance, how do downstream metabolites (as proxies) reduce pleiotropic bias compared to direct genetic instruments? A more detailed explanation of the theoretical framework or a directed acyclic graph (DAG) explicitly mapping the assumed relationships between genetic instruments, target exposure, proxy exposure, and outcome could be preferred.
- (2) A related approach-using downstream biomarkers as proxy exposures-was previously proposed by Schmidt et al. (2020) for drug target validation. To highlight the novelty of the current study, the authors should clarify how their work differs. "Schmidt, A.F., Finan, C., Gordillo-Marañón, M. et al. Genetic drug target validation using Mendelian randomisation. *Nat Commun* 11, 3255 (2020). <https://doi.org/10.1038/s41467-020-16969-0>".
- (3) The authors mention that cell-type-specific gene regulation can introduce pleiotropic effects, but it remains unclear to me how this challenge was addressed in the case studies.
- (4) As the authors mentioned in the manuscript, the proposed framework requires two key assumptions on IVs and the proxy exposure. To assist researchers in applying this method, could the authors provide guidelines on how to select IVs and proxy exposure in practice?

Associate Editor Comments:

We thank the reviewers for the very helpful comments that we believe have greatly improved the clarity of the manuscript. We have made the following major changes to the manuscript:

1. We have added a new section to the manuscript titled “Guidelines for performing *cis*-MR with proxy exposures” and a corresponding Figure 6 to provide more explicit recommendations for performing *cis*-MR with proxy exposures.
2. We have added additional replications for the two HAL missense variants that are one of the main focus of the manuscript. In particular, we now demonstrate that the variants have highly concordant effects on histidine in the Estonian Biobank ($n = 185,352$, Figure S5). We also show that in addition to vitamin D, these variants are also strongly associated with skin cancer risk in the recent meta-analysis of FinnGen, Million Veterans Program and UK Biobank (Figure S7).
3. We provide further evidence that when using plasma histidine as a proxy readout of HAL protein function in skin, it is important to restrict instruments to genetic variants that are likely to have the same effect on HAL protein function in both the trait-relevant context (skin) and the proxy context (plasma). In particular, we demonstrate that using greedy LD pruning around the HAL gene would miss the known causal effect of disrupting HAL protein function on vitamin D levels (Figure S9).

Reviewer #1 :

Review Comments

This paper focuses on the problem of horizontal pleiotropy in the *cis*-MR studies. The authors present two case studies where the metabolites, as downstream phenotypes of target genes' biological processes, can be used as proxy exposures to infer the causal effects of the genes on outcome traits. This method has two advantages: the sample size of metabolite association studies is usually much larger than the eQTL/pQTL studies; and the metabolite as proxy exposure is less affected by horizontal pleiotropy. The authors also highlight the assumptions needed in such proxy-based *cis*-MR studies.

Overall, the manuscript is well-structured, provides biological insights into the usage of proxy-based *cis*-MR, and summarizes the benefits and limitations of this approach, which is valuable for guiding the MR society in using this class of method. I have several comments regarding the clarity of the analysis process and results.

Thank you!

1. For the MR analysis shown in Figure 5B, which variants were used as instruments to obtain the slope -0.152 ? Were all 3 missense variants used as instruments? Was the HAL eQTL variant also included as an instrument?

We have now added Figure S8 in which we included all three missense variants into the MR analysis (with and without the HAL eQTL variant):

Figure S8. MR analysis between HAL function and vitamin D, using all three fine-mapped missense variants as instruments. (A) MR analysis with the three HAL missense variants [12_95977953_C_T (rs61937878), 12_95986106_C_T (rs117991621) and 12_95994812_G_A (rs143854097)] included in the analysis. (B) MR analysis with the HAL eQTL variant added to the three missense variants. Pairwise LD between the variants is shown in Table S2.

We now also discuss the third (discordant) missense variant in more detail in the manuscript (p8, lines 280-290).

“Curiously, our fine-mapping also identified a third missense variant (12_95994812_G_A, rs143854097) in the HAL gene with an even lower allele frequency (MAF ~ 0.1%) that was only associated with histidine level and not with vitamin D level (Figure 5A). This variant was not imputed in the Estonian Biobank, so we could not replicate its effect on plasma histidine in an independent sample. The variant was also not included in the FinnGen+MVP+UKBB meta-analysis for skin cancer, so we could also not assess its effect on skin cancer risk. Given that the rs143854097 variant had a discordant effect relative to the other two missense variants as well as the HAL eQTL variant, we suspect that the lack of association with vitamin D might represent either a technical artefact due to its low allele frequency or an unknown tissue-specific mechanism. For completeness, including the rs143854097 missense variant into the vitamin D MR analysis attenuated the effect size estimate (Figure S8).”

2. How to choose instruments in the proxy-based MR analysis? I understand that using the missense variants is preferred due to their clear relevance to the target gene. In general, if one hopes to apply the proxy-based method, what kind of criteria would be needed to filter the instruments? Can the authors provide a suggested procedure to select valid instruments for proxy exposure?

This is a great point! We have now added a new section to the manuscript titled “Guidelines for performing *cis*-MR with proxy exposures” and a corresponding Figure 6 (see also response to point 1 from Reviewer 2). In Step 3 of the guidelines, we now explicitly cover instrument selection for proxy exposures:

3. As the authors noted, instruments are usually highly correlated in *cis*-MR. For example, two missense variants of *PKLR* gene are used as instruments in the first study and three missense variants of *HAL* gene are used in the second study. What is the correlation among these variants? Is the LD accounted for in the MR analyses?

We have now added Table S2 with the pairwise LD values for the missense variants in the *HAL* and *PKLR* genes. The largest observed r^2 between any two variants was 0.008, indicating that there is no strong LD between the missense variants. As a result, we decided not to account for this (very small LD) in our MR analysis. However, the D' measure of LD was very close to 1 for all tested pairs, likely due to the very low allele frequencies of these variants so that not all possible haplotypes are present in the data.

We have now clarified this in the main text.

On page X, lines YYY:

“The two *PKLR* variants were also not in high linkage disequilibrium (LD) with each other (Table S2).”

We have also added references to Table S2 to the captions of Figures 2 and 5.

4. In the second example, the causal association of interest is the effect of *HAL* activity on Vitamin D, which happens in skin. However, the proxy exposure, histidine, was measured in plasma. How should one interpret the causal effect when the proxy is measured in a tissue different than the target process?

This is a very important point and indeed one of the main novel aspects of our work. The key point is that if one can reasonably assume that the instruments included in the analysis have the same effect on the exposure in the causal context (e.g. skin) and in the proxy context (e.g. plasma) then MR estimates can be interpreted as usual. This assumption is most likely to be satisfied for fine mapped missense and putative loss-of-function variants. However, if the genetic variant effects are different in the two contexts, then this can give highly misleading MR estimates. We have now revised the paragraphs discussing the HAL results to make this point clearer:

Page 11, lines 268-278: *"We hypothesised that for the two HAL missense variants, we could use their effect on reducing plasma histidine level as a proxy measure for their effect on HAL protein function. Using this approach with MR, we detected a significant causal effect of -0.152 between increased HAL protein function (proxied by reduction in plasma histidine) and vitamin D level (Figure 5B). This estimate remained unchanged when histidine was measured in the Estonian Biobank (Tambets et al. 2025) instead of the UK Biobank (beta = -0.163 , Figure S5). Furthermore, using GWAS summary statistics from a skin cancer meta-analysis across the FinnGen, Million Veterans Program and UK Biobank (FinnGen+MVP+UKBB) biobanks ("MVP-Finngen-UKBB meta-analysis") as an outcome, we also detected a proportional effect of these two missense variants on reducing skin cancer risk (beta = -0.192 , Figure S7), mirroring the common HAL eQTL variant associations discussed above (Figure 3C)"*

Page 11-12, lines 292-310: *"An important feature of our MR analysis is that plasma histidine only acts as a proxy readout of variant effect on HAL function (see diagram on Figure 5A) and is unlikely to have a direct causal effect on vitamin D. This raises the question of which histidine-associated variants are valid instruments for causal inference. Most obviously, variants outside the HAL locus should not exhibit consistent effects on vitamin D, which was indeed the case (Figure 5A,C). However, things are also more complicated within the HAL locus. For example, using the skin-specific eQTL variant (rs3819817) with plasma histidine level as exposure would have yielded a highly misleading estimate of $-0.0414/0.014 = -2.96$ (Wald ratio) (Figure 5B). This is because this variant likely has a much larger effect on HAL function in the skin, the causal tissue for vitamin D level, than in the tissues that determine histidine level in plasma. Interestingly, using the expression level of HAL in the skin as the exposure (instead of plasma*

histidine level) with the same instrument yielded a causal effect estimate of $-0.0414/0.49 = -0.084$ (Wald ratio), which aligns more closely with the estimate from the two missense variants (-0.152). Alternatively, using greedy LD pruning to select instruments around the HAL gene (with plasma histidine as exposure) also yielded a null MR estimate (Figure S9), suggesting that regulatory variants could often have discordant effects on HAL function in the skin compared to other tissues that determine plasma histidine levels. Thus, the need to consider tissue- and cell-type-specific effects poses a significant limitation to using variant effects on circulating metabolites (or other molecular traits) as proxy measures for protein function.”

For your convenience, here are the newly added Figures S7 and S9:

Figure S7. MR analysis examining the relationship between proxied HAL protein activity and skin cancer risk. Variant effect on plasma histidine was measured in the UK Biobank and variant effect on skin cancer was extracted from the GWAS meta-analysis of UK Biobank, FinnGen and Million Veterans Program (MVP) cohorts (<https://mvp-ukbb.finnngen.fi/>). The two HAL missense variants included in the analysis are 12_95977953_C_T (rs61937878) and 12_95986106_C_T (rs117991621). LD between the variants is shown in Table S2. The individual betas and standard errors are shown in Table S3.

Figure S9. *cis*-MR at the *HAL* locus with greedy LD pruning to select instruments.

Instruments were selected among variants with MAF > 0.01 using a greedy LD pruning approach ($r^2 < 0.01$). The three *HAL* missense variants were not included due to their low allele frequency. (A) MR analysis using the multiplicative random-effect inverse-variance weighted MR (IVW-MR) method. (B) MR analysis using the MR Locus method.

5. The authors caution the general application of this approach to arbitrary gene regions and proxy exposure pairs. In MR study of a pair of complex traits, one can use established null pairs to benchmark the calibration of a method, e.g., using the hair color as outcome. Is there a parallel approach to calibrate *cis*-MR, which might help rule out the invalid proxy exposures?

We agree that the use of positive and negative controls is a very powerful approach and we now also discuss this in the guidelines, see page 12, lines 392-394:

“Step 4. Use positive and negative control outcomes to validate genetic instruments (Sanderson et al. 2021; Jiesisibieke et al. 2025). Selecting the most appropriate controls for *cis*-MR will likely require extensive domain knowledge.”

However, we believe that the most informative positive and negative controls are likely to be specific to the causal question (and target protein) of interest and thus not easily automatable.

6. Can the estimated causal effects be reproduced when using a different metabolite study?

We have now replicated the *cis*-MR analysis involving *PKLR*, *PFKM* and *HAL* missense variants metabolite GWAS estimates from the Estonian Biobank. For the four variants present in the Estonia Biobank, the estimates results are highly concordant with our original analysis:

Here is the newly added Figure S5:

Figure S5. Replication of the *cis*-MR analysis using metabolite data from the Estonian Biobank. (A) Replication of the *cis*-MR analysis between proxied glycolysis pathway activity and RBC count. Variant effect on pyruvate has now been measured in the Estonian Biobank ($n = 185,352$) while variant effect on RBC count has still been measured in the UK Biobank as in the primary analysis (Figure 2D). The two variants included in the analysis are the *PKLR* missense variant 1_155291918_G_A and the *PFKM* missense variant 12_48118502_C_A. The other two variants were not imputed in the Estonian Biobank due to their low allele frequency. (B) Replication of the *cis*-MR analysis between proxied HAL activity and plasma vitamin D levels. The variant effect on histidine (exposure) has now been measured in the Estonian Biobank ($n = 185,352$) while the variant effect on vitamin D has still been measured in the UK Biobank as in the primary analysis. The two *HAL* missense variants included in the analysis are 12_95977953_C_T (rs61937878) and 12_95986106_C_T (rs117991621). LD between the variants is shown in Table S2.

7. Panels B and D are mislabeled in the caption of Figure 1 (line 53-54). Should be "(D) the same scenario as in C"

Thank you. This has now been fixed.

Reviewer #2 :

The authors propose the use of proxy exposures in *cis*-Mendelian randomization (*cis*-MR) analysis as a strategy to overcome limitations in traditional MR approaches. This method is illustrated through two case studies focusing on glycolysis and vitamin D synthesis pathways. Several key aspects require further clarification:

(1) The study suggests that proxy exposures help minimize horizontal pleiotropy, but the rationale behind this claim is not fully developed. For instance, how do downstream metabolites (as proxies) reduce pleiotropic bias compared to direct genetic instruments? A more detailed

explanation of the theoretical framework or a directed acyclic graph (DAG) explicitly mapping the assumed relationships between genetic instruments, target exposure, proxy exposure, and outcome could be preferred.

We have now added a new section to the manuscript titled “Guidelines for performing *cis*-MR with proxy exposures” and a corresponding Figure 6 to provide more explicit recommendations for performing *cis*-MR with proxy exposures. In particular, Figure 6 contains three DAGs illustrating the possible relationships between target proteins, proxy readouts and whether the proxy readout is likely to be on the causal path from the target exposure to the outcome.

We have copied the newly added section and Figure 6 below:

Guidelines for performing *cis*-MR with proxy exposures

*To make it easier to perform *cis*-MR studies with proxy exposures and avoid some of the common pitfalls, we have put together the following guidelines also illustrated on Figure 6:*

Step 1. *Familiarise yourself with the best practices and pitfalls of *cis*-MR and drug target MR. Some excellent places to start are (Burgess et al. 2019; Schmidt et al. 2020; Gill et al. 2021; Sanderson et al. 2022; Gill et al. 2024) and references contained therein.*

Step 2. *Assess whether the target protein has a plausible proximal, downstream readout that reflects its biological function.*

Step 3. *Assess if the proxy exposure is likely to be on the causal pathway from the protein to the outcome of interest (e.g. LDL for coronary artery disease) or if it is measured in a non-causal context (e.g. plasma histidine for vitamin D). As highlighted by our HAL case study, proxy exposures measured in non-causal contexts might require restricting instrument selection to variants that have the same effect on protein function in all contexts (e.g. missense and putative loss-of-function variants).*

Step 4. *Use positive and negative control outcomes to validate genetic instruments (Sanderson et al. 2021; Jiesisibieke et al. 2025 Jun 9). Selecting the most appropriate controls for *cis*-MR will likely require extensive domain knowledge.*

Step 5. *Recognize that nearly all protein function readouts (e.g. gene or protein abundance) are proxies and may not capture true function due to technical artefacts or cell-type mismatch.*

Step 6. *Understand that no checklist can guarantee robust causal inference. Sometimes the most important insight might be realising when MR should not be performed (Jiesisibieke et al. 2025 Jun 9; Reed et al. 2025).*

Guidelines for performing *cis*-MR analysis with proxy exposures

STEP 1 Familiarise yourself with the best practices and pitfalls of *cis*-MR and drug target MR.

STEP 2 Assess whether the target protein has a plausible proximal, downstream readout that reflects its biological function. **If no such readout exists, or if the genetic instruments are not strongly associated with it, *cis*-MR should not be pursued.**

STEP 3 To select the genetic instruments, determine whether the proxy exposure (**A**) is on the causal pathway from the protein to the outcome (vertical pleiotropy), or (**B**) is measured in a non-causal context.

STEP 4 Use positive and negative control outcomes to validate the genetic instruments.

STEP 5 Recognize that nearly all protein function readouts (e.g., gene or protein abundance) are proxies and may not fully capture true function due to cell-type mismatch or weak correlation.

STEP 6 Understand that no checklist guarantees robust causal inference. Reliable conclusions require critical evaluation and triangulation of evidence from multiple sources. **Importantly, know when MR is not appropriate despite available data.**

Figure 6. Guidelines for performing *cis*-MR analysis with proxy exposures.

To be clear, the reduction in horizontal pleiotropy comes from focussing the analysis to specific *cis* regions around the target protein of interest (i.e. conducting *cis*-MR) rather than performing genome-wide MR using all genome-wide significant variants for a given metabolite. This is well established in the literature and not our novel claim (see e.g. reviews by Burgess et al, 2019; Gill et al, 2021 that we now also cite in Step 1 of our guidelines. Similarly, using the most direct

readout of the variant effect on protein function (e.g. enzymatic activity) is preferable, but this might not always be available. In that scenario, we might want to use a reasonable proxy readout. Our manuscript explores the challenges and opportunities when performing *cis*-MR analysis with proxy readouts of protein function.

(2) A related approach-using downstream biomarkers as proxy exposures-was previously proposed by Schmidt et al. (2020) for drug target validation. To highlight the novelty of the current study, the authors should clarify how their work differs.

"Schmidt, A.F., Finan, C., Gordillo-Marañón, M. et al. Genetic drug target validation using Mendelian randomisation. *Nat Commun* 11, 3255 (2020). <https://doi.org/10.1038/s41467-020-16969-0>".

Thank you for the reference. We now refer to it in multiple places in the manuscript, including at the end of paragraph 6 of the introduction (p5, lines 124-126):

"However, a systematic analysis of when and how these proxy measures for gene or protein function can be used for causal inference is still lacking, especially outside of the lipids-cardiovascular domain (Schmidt et al. 2020)."

We believe our main novelty lies in expanding the use of proxy exposures beyond lipid and cardiovascular traits (as encouraged by Schmidt *et al.* 2020 in in their Discussion), In particular, we directly explore what happens when the proxy readout is measured in a non-causal context (e.g. HAL protein activity in skin proxied by histidine level in plasma) and highlight the importance of only using genetic instruments that are likely to have the same effect on the target exposure and its proxy readout (e.g. missense or loss-of-function variants). We now also explicitly highlight this in the newly added guidelines as well as in Figure 6 (see Step 3).

(3) The authors mention that cell-type-specific gene regulation can introduce pleiotropic effects, but it remains unclear to me how this challenge was addressed in the case studies.

We have now rewritten the part of the manuscript describing the HAL results to further clarify the importance of considering cell-type specific effects (pages 11-12, lines 292-310):

"An important feature of our MR analysis is that plasma histidine only acts as a proxy readout of variant effect on HAL function (see diagram on Figure 5A) and is unlikely to have a direct causal effect on vitamin D. This raises the question of which histidine-associated variants are valid instruments for causal inference. Most obviously, variants outside the HAL locus should not exhibit consistent effects on vitamin D, which was indeed the case (Figure 5A,C). However, things are also more complicated within the HAL locus. For example, using the skin-specific eQTL variant (rs3819817) with plasma histidine level as exposure would have yielded a highly

misleading estimate of $-0.0414/0.014 = -2.96$ (Wald ratio) (Figure 5B). This is because this variant likely has a much larger effect on HAL function in the skin, the causal tissue for vitamin D level, than in the tissues that determine histidine level in plasma. Interestingly, using the expression level of HAL in the skin as the exposure (instead of plasma histidine level) with the same instrument yielded a causal effect estimate of $-0.0414/0.49 = -0.084$ (Wald ratio), which aligns more closely with the estimate from the two missense variants (-0.152). Alternatively, using greedy LD pruning to select instruments around the HAL gene (with plasma histidine as exposure) also yielded a null MR estimate (Figure S9), suggesting that regulatory variants could often have discordant effects on HAL function in the skin compared to other tissues that determine plasma histidine levels. Thus, the need to consider tissue- and cell-type-specific effects poses a significant limitation to using variant effects on circulating metabolites (or other molecular traits) as proxy measures for protein function.

We now also discuss this in Step 3 of the guidelines (and Figure 6):

“Step 3. Assess if the proxy exposure is likely to be on the causal pathway from the protein to the outcome of interest (e.g. LDL for coronary artery disease) or if it is measured in a non-causal context (e.g. plasma histidine for vitamin D). As highlighted by our HAL case study, proxy exposures measured in non-causal contexts might require restricting instrument selection to variants that have the same effect on protein function in all contexts (e.g. missense and putative loss-of-function variants).”

(4) As the authors mentioned in the manuscript, the proposed framework requires two key assumptions on IVs and the proxy exposure. To assist researchers in applying this method, could the authors provide guidelines on how to select IVs and proxy exposure in practice?

We have now added the guidelines as outlined in response to point 1.

September 9, 2025

GENETICS-2025-308511

Mendelian randomisation with proxy exposures: challenges and opportunities

Dear Dr. Alasoo:

Two experts in the field have reviewed your manuscript, and I have read it as well. I am pleased to inform you that, with minor revisions, it is potentially suitable for publication in GENETICS. The reviewers have comments and concerns that need to be addressed in a revised manuscript. You can read their reviews at the end of this email.

We look forward to receiving your revised manuscript with code. This will ensure that readers or users can reproduce your results, and make your methods more applicable. Please let the editorial office know approximately how long you expect to need for revisions.

Upon resubmission, please include:

1. A clean version of your manuscript;
2. A marked version of your manuscript in which you highlight significant revisions carried out in response to the major points raised by the editor/reviewers (track changes is acceptable if preferred);
3. A detailed response to the editor's/reviewers' comments and to the concerns listed above. Please reference line numbers in this response to aid the editors.

Additionally, please ensure that your resubmission is formatted for GENETICS.

<https://academic.oup.com/genetics/pages/general-instructions>

Follow this link to submit the revised manuscript: Link Not Available

Sincerely,

Can Yang
Associate Editor
GENETICS

Approved by:
Hongyu Zhao
Senior Editor
GENETICS

Reviewer #1 :

The authors have addressed my major concerns. I appreciate their efforts in providing a pipeline of proxy-based MR inference. It would help others to more easily implement this procedure if the authors could provide the code to reproduce their results.

Reviewer #2 :

The authors have addressed all my questions. I have no more comments.

Associate Editor Comments:

Reproducibility is critical. The authors should have well organized code for reproducing their results.

Dear Dr. Yang,

We have now uploaded the relevant software code to a GitHub repository (https://github.com/AlasooLab/MR_with_proxy_exposures) and updated the Data availability statement to include a link to this repository:

Data availability

The GWAS summary statistics for the 56 metabolic traits are available from Zenodo (<https://doi.org/10.5281/zenodo.13821209>). The fine-mapped credible sets and log Bayes factors from SuSiE are available from Zenodo (<https://doi.org/10.5281/zenodo.13821038>).

The UK Biobank summary statistics and fine-mapping results for RBC count and vitamin D (Kanai et al. 2021) were downloaded from Google Cloud (link). The GWAS summary statistics for skin cancer from (Seviiri et al. 2022) were downloaded from the GWAS Catalog (accession GCST90137411). The skin cancer summary statistics from the FinnGen-MVP-UKBB meta-analysis (phenotype C3_OTHER_SKIN_EXALLC) were downloaded from the FinnGen website (<https://mvp-ukbb.finnngen.fi/about>). The GWAS summary statistics for “childhood sunburn occasions” (UK Biobank data field 1737) were downloaded from the Neale lab website (https://github.com/Nealelab/UK_Biobank_GWAS). The histidine and pyruvate GWAS summary statistics from the Estonian Biobank were downloaded from the GWAS Catalog (accessions GCST90449390 and GCST90449502).

Data analysis code is available from GitHub (https://github.com/AlasooLab/MR_with_proxy_exposures). The GWAS and fine-mapping Nextflow workflow is available from GitHub (<https://github.com/AlasooLab/reGSusie>). All forest plots were made with the ggforestplot R package (<https://github.com/NightingaleHealth/ggforestplot>).

We have now also formatted the manuscript for GENETICS.

Kind regards,
Kaur Alasoo

September 17, 2025
RE: GENETICS-2025-308594

Dr. Kaur Alasoo
Tartu Ulikool
Institute of Computer Science
Narva street 18
Tartu
Estonia

Dear Dr. Alasoo:

Congratulations, your manuscript titled "Mendelian randomisation with proxy exposures: challenges and opportunities" is accepted for publication in GENETICS! Many thanks for submitting your research to the journal.

To Proceed to Publication:

1. Format your article according to GENETICS style: <https://academic.oup.com/genetics/pages/author-guidelines>
2. Ensure that you comply with data and community resource citation guidelines:
<https://academic.oup.com/genetics/pages/author-guidelines#section-5-9-2>
3. Upload your final files at <https://genetics.msubmit.net>
4. Add oupsupport@scipris.com and genetics.oup@novatechset.com (or the domains @scipris.com and @novatechset.com) to your email program's "safe senders" list. You will be contacted by both at various points during the production process.

Notes:

- Your currently-accepted manuscript (unedited, as submitted, reviewed, and accepted) will be published at GENETICS and deposited into PubMed as an Advance Access article. Notify sourcefiles@thegsajournals.org before signing your license if you do not wish to publish your article via Advance Access.
- We invite you to submit an original color figure related to your paper for consideration as cover art. Please email your submission to the editorial office or upload it with your final files. You can submit a small-sized image for evaluation, and if selected, the final image must be a TIFF file 2513px wide by 3263px high (8.375 by 10.875 inches; resolution of 600ppi). Please avoid graphs and small type.
- After files are sent to Oxford University Press we use SciPris to manage article licensing and payment. If you do not have a SciPris account, you will receive an email from no-reply@scipris.com to sign up to use Oxford University Press' author portal. After logging in, follow the online instructions to sign your license and arrange any payment due.

If you have any questions or encounter any problems while uploading your accepted manuscript files, please email the editorial office at sourcefiles@thegsajournals.org.

Sincerely,

Can Yang
Associate Editor
GENETICS

Approved by:
Hongyu Zhao
Senior Editor
GENETICS

Review comments (if applicable):

Reviewer #1 :

I have no further comments

Reviewer #2 :

I have no further questions!